# In Vitro and In Silico Analyses of New Cinnamid and Rosmarinic Acid-Derived Compounds Biosynthesized in *Escherichia coli* as *Leishmania amazonensis* Arginase Inhibitors

**DOI:** 10.3390/pathogens11091020

**Published:** 2022-09-07

**Authors:** Julio Abel Alfredo dos Santos Simone Come, Yibin Zhuang, Tianzhen Li, Simone Brogi, Sandra Gemma, Tao Liu, Edson Roberto da Silva

**Affiliations:** 1Departamento de Pré-Clínicas, Faculdade de Veterinária, Universidade Eduardo Mondlane, Av. de Moçambique, Km 1.5, Maputo CP 257, Mozambique; 2Tianjin Institute of Industrial Biotechnology, Chinese Academy of Sciences, Tianjin 300308, China; 3Key Laboratory of Systems Microbial Biotechnology, Chinese Academy of Sciences, Tianjin 300308, China; 4Department of Pharmacy, University of Pisa, Via Bonanno 6, 56126 Pisa, Italy; 5Department of Biotechnology, Chemistry and Pharmacy, University of Siena, Via Aldo Moro 2, 53100 Siena, Italy; 6Laboratório Farmacologia e Bioquímica (LFBq), Departamento de Medicina Veterinária, Universidade de São Paulo Faculdade de Zootecnia e Engenharia de Alimentos, Pirassununga 13635-900, SP, Brazil

**Keywords:** arginase, *Leishmania amazonensis*, biosynthesis, natural products, molecular modeling

## Abstract

Arginase is a metalloenzyme that plays a central role in *Leishmania* infections. Previously, rosmarinic and caffeic acids were described as antileishmanial agents and as *Leishmania amazonensis* arginase inhibitors. Here, we describe the inhibition of arginase in *L. amazonensis* by rosmarinic acid analogs (**1–7**) and new caffeic acid-derived amides (**8–10**). Caffeic acid esters and amides were produced by means of an engineered synthesis in *E. coli* and tested against *L. amazonensis* arginase. New amides (**8–10**) were biosynthesized in *E. coli* cultured with 2 mM of different combinations of feeding substrates. The most potent arginase inhibitors showed Ki(s) ranging from 2 to 5.7 μM. Compounds **2–4** and **7** inhibited *L. amazonensis* arginase (L-ARG) through a noncompetitive mechanism whilst compound **9** showed a competitive inhibition. By applying an in silico protocol, we determined the binding mode of compound **9**. The competitive inhibitor of L-ARG targeted the key residues within the binding site of the enzyme, establishing a metal coordination bond with the metal ions and a series of hydrophobic and polar contacts supporting its micromolar inhibition of L-ARG. These results highlight that dihydroxycinnamic-derived compounds can be used as the basis for developing new drugs using a powerful tool based on the biosynthesis of arginase inhibitors.

## 1. Introduction

Leishmaniasis is a neglected tropical disease with a public health importance. Cutaneous leishmaniasis has affected 1 million people in the last 5 years; if untreated, visceral leishmaniasis can result in more than 20,000 deaths each year [1]. Unfortunately, there is a lack of effective drugs for the treatment of leishmaniasis and the few available therapeutic options present a high toxicity associated with treatment resistance. The most antileishmanial agents are highly toxic injectable medicines such as pentavalent antimonials (sodium stibogluconate and meglumine antimoniate) and liposomal amphotericin B. The resistance of these drugs has also been described for both classes as well for miltefosine, an oral medicine used to treat leishmaniasis [2].

The evolution of leishmaniasis depends on the balance among the Th1 cytokines (which trigger the activation of the L-arginine metabolic pathways for nitric oxide (NO) production, which is responsible for killing the parasite) and Th2 cytokines (which determine the induction of the host arginase and enhance the *Leishmania* infection) [3]. Arginase is an enzyme that catalyzes the hydrolysis of L-arginine to L-ornithine and urea. In *Leishmania*, the arginase enzyme plays a crucial role in polyamine precursor metabolism, as highlighted through the use of a null arginase mutant obtained by a targeted gene replacement [4,5]. Polyamines are substrates for the production of trypanothione, an antioxidant that neutralizes reactive oxygen species (ROS) and NO [6,7]. Arginase activity deprives the nitric oxide synthase 2 (NOS2) of the substrate (arginine), decreasing NO biosynthesis in the host defense cells and providing polyamines that can increase the parasite proliferation [3].

Due to the relevant action in the L-arginine metabolism pathway, arginase may be an important target for leishmaniasis treatments. Arginase inhibitors were shown to reduce the parasite burden in *Leishmania*-infected BALB/c mice [8]. We previously identified compounds with catechol groups [9,10,11] and thiosemicarbazide [12] as crucial *L. amazonensis* arginase (L-ARG) inhibitors. The compound verbascoside, a major constituent of the medicinal plant *Stachytarpheta cayennensis* that is used to treat leishmaniasis in Brazil [13] and malaria in Peru [14], inhibits L-ARG and it is active against both the promastigote and intracellular amastigotes of the parasite [15,16]. Other natural dihydroxycinnamic compounds (caffeic acid, chlorogenic acid, and rosmarinic acid) have shown in vitro and in vivo activities against *L. amazonensis* [17,18] as well as an inhibitory profile against L-ARG [19]. Based on the antileishmanial activity of dihydroxycinnamic compounds [18] and L-ARG inhibition, we tested compounds previously produced via engineered *E. coli* [20,21]. In addition, we synthesized three new amides containing the 3,4-dihydroxycinnamic moiety and tested them against L-ARG. The small molecules showed great potential in targeting L-ARG, which could be used to guide the development of new drugs against leishmaniasis. 

Furthermore, the competitive inhibitor, compound **9**, was extensively investigated by computational studies to gain an insight into the mechanism of action of this derivative in inhibiting the L-ARG enzyme. Accordingly, by using a computer-based protocol based on molecular docking and molecular dynamics (MD), we comprehensively assessed the behavior of compound **9** as a competitive inhibitor of the mentioned enzyme.

Scaling up the production of rosmarinic acid and other compounds using *E. coli* could improve the feasibility of using these compounds in animal studies and clinical trials. Additionally, engineered *E. coli* provides a large avenue to explore the molecular modifications that alter donor and acceptor substrates to synthesize new potential antileishmanial agents targeting the L-ARG enzyme.

## 2. Results

### 2.1. Production of Compounds 8–10 in E. coli via Feeding Experiments

In addition to the previously synthesized compounds **1–7** (Table 1), three new amides (**8–10**) were biosynthesized in *E. coli* cultured at 30 °C for 48 h and supplemented with 2 mM of different combinations of feeding substrates (2-amino-5-methylbenzoic acid, 2-amino-4,5-difluorobenzoic acid, and 2-amino-4-chlorobenzoic acid as acceptors; *trans*-4-hydroxycinnamic acid, 3-(4-hydroxyphenyl)propionic acid, and *trans*-3,4-dihydroxycinnamic acid as donors) (Figure 1). The compounds were characterized using NMR and MS.

### 2.2. Arginase Inhibition

Five biosynthetic compounds containing a catechol group, which were previously synthesized via engineered *E. coli* [20,21], showed L-ARG inhibition (Table 1) with IC_50_ values that ranged from 1.9 μM (**4**) to 36.2 μM (**7**). Compounds that lacked the catechol group were inactive (**1**, **6**, **8**, and **10**) or showed a weak inhibition (**5**) of L-ARG. The new compound **9** showed an IC_50_ value of 5.5 ± 0.5 µM and was a unique cinnamide that was active against L-ARG (Table 1). Compound **9** was synthesized using 3,4-dihidroxycinnamic acid as a donor substrate and 2-amino-4,5-difluorobenzoic as an acceptor in the *E. coli* system, which generated an unnatural difluorobenzoic cinnamide with a relevant L-ARG inhibitory profile. 

### 2.3. Kinetics of Arginase Inhibition 

The enzyme kinetics were performed to determine the mechanism of inhibition and the dissociation constant (Ki) of the compounds via the simultaneous analysis of the Dixon [22] and Cornish-Bowden [23] plots. The Ki inhibition constants (complex EI), which refer to the equilibriums established between the enzyme (E) and substrate (S) in the presence of an inhibitor (I), were determined (Table 1). The rosmarinic acid analogs (**2–4**) showed a noncompetitive mechanism considering L-ARG inhibition. All slopes were significantly different (*p* ≤ 0.05). The interception point was used to calculate the Ki values and to obtain the mechanism of inhibition via the graph method; this result was in agreement with the previous results of an invariable IC_50_ obtained for three concentrations of the substrates (Figure 1). The new compound **9** was found to be a competitive L-ARG inhibitor (Table 1).

### 2.4. Computational Studies

The binding mode of the competitive inhibitor, compound **9**, was investigated applying a computational protocol based on molecular docking calculations and an MD simulation with an evaluation of the ligand binding energy as previously reported by us [19,24,25]. In general, we preferred to investigate only the binding of the competitive inhibitors of L-ARG because, for the other inhibitors, the mechanism is not completely understood and a discussion on the binding of these compounds to L-ARG could be extremely speculative. Accordingly, considering molecular docking studies, we conducted computational experiments employing Glide and Prime software and focusing attention on the substrate binding site using a crystal structure of L-ARG belonging to *Leishmania mexicana* (PDB ID 4IU1). The output of this calculation is reported in Figure 2. 

Compound **9** was able to interact with the L-ARG binding site by a series of polar and hydrophobic interactions. In particular, the catechol moiety was involved in a metal coordination bond by one of its hydroxyl groups. Moreover, the hydroxyl groups could form two H-bonds with side chains of Thr257 and Glu288. Furthermore, the aromatic component of the catechol moiety established a double π-π stacking with His139 and His154. Additionally, the carboxylic function belonging to the difluorobenzoate moiety was able to form a strong H-bond network by interacting with Thr148 (side chain), Val149 (backbone), and Ser150 (side chain and backbone). This pattern of interaction accounted for a docking score and estimated ligand binding energy (XP score = −8.51 kcal/mol; ΔG_bind_ = −78.29 kcal/mol, respectively) in line with our previous computational studies conducted on comparable compounds and in line with the micromolar inhibition of the enzyme.

In order to validate the docking output and to assess the stability of the binding mode of compound **9** retrieved by the molecular docking calculation and related timeline behavior, we conducted a 300 ns MD simulation study in an explicit solvent on the biological system composed of L-ARG/compound **9**. The resultant trajectory was exhaustively analyzed using different standard simulation parameters, including a root mean square deviation (RMSD) assessment for each backbone atom and ligand and a root mean square fluctuation (RMSF) of each protein residue (Figure 3). 

The selected complex showed general stability from the early stages of the MD simulation, as indicated by the results found by calculating the RMSD without major expansion and/or contraction events of the examined system during the entire simulation after the binding of compound **9**. We did not observe a significant variation in the RMSD (Figure 3A). This stability was also substantiated by observing the RMSF calculated for the mentioned complex. The RMSF indicated the difference between the atomic Cα coordinates of the protein from its average position during the MD simulation. This calculation was helpful to characterize the flexibility of individual residues in the protein backbone. The considered system did not show significant fluctuation phenomena, with the exclusion of a restricted number of residues in the N- and C-terminal regions of L-ARG (Figure 3B). In contrast, the conformational alterations of the critical residues of the L-ARG binding cleft (lowest RMSF value) confirmed the capacity of the compounds to form stable interactions within the protein. To gain a further insight into the behavior of compound **9** in the L-ARG binding site, we performed a comprehensive analysis of the MD simulation, examining the main interactions established by compound **9** in the L-ARG active site. The result of the evaluation regarding the mentioned complex is reported in Figure 4. In general, observing the trajectory of the MD simulation, compound **9** maintained the interactions found from the molecular docking calculation, targeting the backbone of Thr148 and Val149 as well as the sidechain of Ser150 and Glu188. Furthermore, during the MD simulation, we observed further additional contacts with the backbone of Asp141 and a water-mediated H-bond with Asn152. Moreover, a series of ionic interactions was observed among the catechol moiety and the residues of the enzyme able to coordinate the metal ions (His114, Asp137, Asp243, and Asp245). All these contacts were relevant in stabilizing the discussed binding mode as well as to perturbate the activity of the enzyme. Overall, the MD simulation outcome undeniably validated the significant interactions of compound **9** found by the docking studies, indicating that the compound could behave as an effective inhibitor of L-ARG. This timeline behavior was also highlighted by the dynamic interaction diagram obtained from the 300 ns of the MD simulation as reported in Figure 5.

Finally, we assessed the timeline interaction of compound **9** with the metal reactive center of the L-ARG enzyme (Figure 6). Starting from the complex L-ARG/compound **9** obtained by docking, we observed only one potential metal coordination bond, occurring between the oxygen atom—namely, O1—and the metal ion MN401. As expected, the distance (D1) between the mentioned atoms remained favorable for the duration of the simulation for forming a metal coordination. The distance D2 between the oxygen atom O1 and the metal MN402 appeared not to be favorable for establishing a metal coordination (over 3 Å) and we observed only a sporadic metal coordination bond during the 300 ns of the MD simulation. Surprisingly, the other oxygen atom belonging to the catechol moiety—namely, O2—was able, after the stabilization of the complex, to find a suitable position for establishing an additional metal coordination bond with the other metal ion MN401, considering that from 20 ns of the MD simulation, the distance (D3) was favorable to coordinate the metal ion (Figure 6). The same phenomenon was observed considering the same oxygen atom (O2) and the other metal ion MN402. After an initial stabilization of the bioactive conformation, atom O2 could reach a favorable distance (D4) to interact with the metal ion MN402 for a relevant segment of the MD simulation. Accordingly, the analysis highlighted the capability of compound **9** to appropriately interact within the L-ARG binding site targeting the key residues as well as its ability to form metal coordination bonds with the metal reactive center of the enzyme, indicating the ability to inhibit the L-ARG enzyme, as found by in vitro tests.

## 3. Discussion

Several insights could be highlighted via the analysis of the six compounds that were active against L-ARG and the results were compared with those obtained using a previous model of mammalian arginase [26]. Donor substrates that contained catechol were substituted with a donor containing a phenyl moiety (*trans*-4-dihydroxycinnamic acid or 3-(4-hydroxyphenyl)propionic acid) in compounds **1**, **5**, and **6**; for these compounds, the inhibition of the enzyme decreased at least 100 times (compound **4** vs. **5**) or resulted in inactive compounds (**1** and **6**).

These data highlight the importance of the 3,4-dihydroxyphenyl moiety in a donor substrate to synthesize *L. amazonensis* arginase inhibitors. Compounds **2**–**4** were synthesized using a common donor substrate (3,4-dihydroxyphenyl-propanoic acid) and by varying the acceptor (4-hydroxyphenyllactic acid (HPL), phenyllatic acid (PL), and 3,4-dihidroxyphenyllactic acid (DHPL)); the analysis of the L-ARG inhibition by these compounds showed that the best arginase inhibition was obtained when the acceptor used was HPL. *Leishmania* regulates the gene expression of cationic transporters and increases the uptake of L-arginine [27]; this could be a possible resistance mechanism if the mixed inhibitor of arginase is used as an antileishmanial agent. Therefore, a noncompetitive arginase inhibitor would be a better choice as an antileishmanial drug candidate. The replacement of hydroxyphenyl from compound **4** (IC_50_ 1.9 µM) to the carboxylate moiety in compound **7** (IC_50_ 36.2 µM) decreased the IC_50_ approximately 20 times. Finally, the compounds that were obtained using the same donor (*trans*-3,4-dihydroxycinnamic acid) showed that the difluorobenzoic moiety in the acceptor, which was used to obtain the cinnamide (**9**), exhibited a better activity than the succinic moiety (**7**), which was used for the L-ARG inhibition. 

Recently, a study of natural compounds isolated from *Pluchea carolinensis* [18] showed that caffeic acid (3,4-dihydroxycinnamic) and its derivative compounds chlorogenic acid, ferulic acid, and rosmarinic acid are active against the promastigotes and amastigotes of *L. amazonensis*. Ferulic, rosmarinic, and caffeic acids were effective in reducing the lesion size and parasite burden in a BALB/c mice model of cutaneous leishmaniasis with *L. amazonensis* [18]. 

The in silico investigation, employing a computational protocol based on molecular docking coupled to a ligand binding energy estimation and MD simulation, was demonstrated by us to be a useful approach for a reliable prediction of the putative binding modes and affinity of the competitive inhibitors for L-ARG. In this case, for compound **9**, acting as a competitive inhibitor of the enzyme, we observed its ability to target key residues within the binding site of the enzyme. The catechol moiety was projected toward the active site metal ions and the compound could establish a metal coordination bond. In addition, with the same moiety, compound **9** was able to target two additional residues in the reactive metal pocket. The other portion of the molecule by a strong H-bond network could stabilize the retrieved binding mode by interacting with a series of residues located at the entrance of the catalytic cleft. This binding mode of compound **9** supported its low micromolar inhibition of L-ARG. Furthermore, we confirmed the docking results by an extensive MD simulation experiment in which we observed that the binding mode of compound **9** was maintained during 300 ns of an MD simulation in an explicit solvent. 

## 4. Materials and Methods

### 4.1. Materials

CHES (2-(cyclohexylamino) ethane-sulfonic acid) and L-arginine were purchased from Sigma-Aldrich and the reagents for the urea analysis were purchased from Quibasa (Belo Horizonte, MG, Brazil). Compounds **1–7** were obtained via engineered synthesis in *E. coli* [20,21] at the Tianjin Institute of Industrial Biotechnology. Compounds 2-amino-4,5-difluorobenzoic acid, 2-amino-4-chlorobenzoic acid, *trans*-4-hydroxycinnamic acid, 3-(4-hydroxyphenyl)propionic acid, and *trans*-3,4-dihydroxycinnamic acid were purchased from Aladdin Chemistry Co., Ltd. (Shanghai, China). 

### 4.2. Bacterial Strain, Cultivation, and Chemicals

The codon-optimized hydroxycinnamoyl/benzoyl-CoA:anthranilate N-hydroxycinnamoyl/benzoyl transferase (HCBT) was synthesized for its optimal expression in *E. coli*. The 4-Coumarate CoA ligase gene (*At4CL*) was amplified via PCR from the cDNA of *Arabidopsis thaliana.* PET28a-HCBT was constructed by inserting HCBT into PET28a using the restriction sites *Nde* I and *BamH* I; PCDFDuet-At4CL was constructed by inserting At4CL into PCDFDuet using the restriction sites *Nco* I and *BamH* I. Plasmids PET28a-HCBT and PCDFDuet-At4CL/pET28a and PCDFDuet were co-transformed into *E*. *coli* BL21 (DE3), generating the recombinant strain S1 and a negative control strain S2, respectively. 

The strains S1 and S2 were cultivated in liquid Luria–Bertani (LB) broth or on LB agar plates at 37 °C with 50 μg/mL kanamycin and 200 μg/mL streptomycin to maintain the plasmids. For the production of the biotransformation products, 1 mL of the overnight-cultured single colonies of the engineered *E. coli* strain BLRA1 was diluted with 50 mL of a fresh LB medium and incubated at 37 °C at 200 rpm. When the OD_600_ of the culture reached 0.6–0.8, 0.1 mM isopropyl-β-D-thiogalactoside (IPTG) was added to induce the recombinant protein expression at 16 °C for 12 h. Subsequently, the cells were centrifuged, washed, and resuspended in 50 mL of a slightly modified M9 medium (1 × M9 minimal salts, 5 mM MgSO_4_, 0.1 mM CaCl_2_, and 2% (*w/v*) glucose supplemented with 1% (*w/v*) yeast extract). The cells were treated with different feeding substrates at a concentration of 2 mM (2-amino-5-methylbenzoic acid, 2-amino-4,5-difluorobenzoic acid, and 2-amino-4-chlorobenzoic acid as acceptors; *trans*-4-hydroxycinnamic acid, 3-(4-hydroxyphenyl)propionic acid, and *trans*-3,4-dihydroxycinnamic acid as donors). The cells were cultured at 30 °C for 48 h. All substrates were purchased from Aladdin Chemistry Co., Ltd. (Shanghai, China).

### 4.3. Extraction, Isolation, and Identification of the Biotransformation Products

To extract the biotransformation products from the broth, the fermentation was scaled up to 500 mL. After the fermentation broths were centrifuged, the supernatants containing the biotransformation products were extracted using glass columns wet-packed with macroporous resin SP825L (100 mL; Sepabeads, Kyoto, Japan). An aliquot (200 mL) of distilled water and 80% (*v/v*) ethanol were sequentially loaded into the column and were eluted at a constant flow rate of 1 mL/min. An eluate of 80% (*v/v*) ethanol was separately condensed under a reduced pressure. The residue was then dissolved in 2 mL of methanol and purified via semipreparative HPLC using a Shimadzu LC-6 AD with an SPD-20A detector equipped with a YMC-pack ODS-A column (10 × 250 mm; i.d., 5 μm; YMC, Kyoto, Japan). The flow rate was 4 mL/min; the other HPLC conditions were the same as described above. The compounds were tested using NMR and MS. The ^1^H-NMR spectra were analyzed on a Bruker Avance III spectrometer at 400 MHz in CD_3_OD. The chemical shifts were expressed as δ (ppm) and the coupling constants (*J*) were expressed as hertz (Hz). The MS spectra were performed on a Bruker microQ-TOF II mass spectrometer (Bruker BioSpin, Switzerland) equipped with an electrospray ionization (ESI) interface. The purity of the compounds used to test the activity was more than 90% and the compounds were characterized by NMR and MS spectra (Appendix A).

Compound **8**: ^1^H-NMR (CD_3_OD, 400 MHz), δ H 8.83 (d, *J* = 2.1 Hz, 1H), 8.07 (d, *J* = 8.6 Hz, 1H), 7.63 (d, *J* = 15.6 Hz, 1H), 7.50 (d, *J* = 8.6 Hz, 2H), 7.14 (dd, *J* = 8.6, 2.1 Hz, 1H), 6.83 (d, *J* = 8.6 Hz, 2H), 6.52 (d, *J* = 15.6 Hz, 1H); ^13^C-NMR (CD_3_OD, 100 MHz) δ C 170.9, 167.2, 161.2, 144.2, 143.8, 141.0, 133.9, 131.1, 127.3, 123.7, 121.0, 118.8, 116.8, 115.9; HRESI-MS *m*/*z* 316.0382 [M-H]^−^ (calcd. for C_16_H_11_ClNO_4_ 316.0377). 

Compound **9**: ^1^H-NMR (CD_3_OD, 400 MHz), δ H 8.73 (m, 1H), 7.97 (m, 1H), 7.57 (d, *J* = 15.6 Hz, 1H), 7.09 (d, *J* = 2.0 Hz, 1H), 7.00 (dd, *J* = 8.2, 2.0 Hz, 1H), 6.81 (d, *J* = 8.2 Hz, 1H), 6.46 (d, *J* = 15.6 Hz, 1H); ^13^C-NMR (CD_3_OD, 100 MHz) δ C 169.9, 167.7, 149.5, 146.8, 144.6, 127.8, 122.9, 120.9, 120.7, 118.6, 116.5, 115.2, 110.4, 110.1; HRESI-MS *m*/*z* 334.0553 [M-H]^−^ (calcd. for C_16_H_10_F_2_NO_5_ 334.0527).

Compound **10**: ^1^H-NMR (CD_3_OD, 400 MHz), δ H 8.69 (d, *J* = 2.2 Hz, 1H), 8.03 (d, *J* = 8.6 Hz, 1H), 7.12 (dd, *J* = 8.6, 2.2 Hz, 1H), 7.06 (d, *J* = 8.6 Hz, 1H), 6.70 (d, *J* = 8.5 Hz, 1H), 2.94 (t, *J* = 7.6 Hz, 1H), 2.69 (t, *J* = 7.6 Hz, 1H); ^13^C-NMR (CD_3_OD, 100 MHz) δ C 173.7, 170.6, 156.8, 143.3, 141.0, 133.8, 132.5, 130.3, 123.8, 120.9, 116.3, 115.8, 41.5, 31.7; HRESI-MS *m*/*z* 318.0570 [M-H]^−^ (calcd. for C_16_H_13_ClNO_4_ 318.0533). 

### 4.4. Arginase Inhibition and Kinetics

Recombinant L-ARG was expressed and purified as previously described [9]. A stock solution of 70 mM was prepared in DMSO and was stored at −20 °C for each compound just before the experiment. The concentration that inhibited 50% of the catalytic activity of the enzyme (IC_50_) was determined using an inhibitor concentration from 200 μM to 0.05 μM; the concentrations were obtained via a serial dilution in water (1:4). The kinetics were determined using three concentrations of the substrate and three concentrations of inhibitor, as previously described [11]. Two independent experiments were performed in triplicate with a coefficient of nonlinear regression R^2^ ≥ 0.85. The sigmoidal model (log IC_50_) was used to determine the IC_50_ using GraphPad Prism 7 software for Windows (San Diego, CA, USA).

### 4.5. Computational Details

*Molecule preparation*: The three-dimensional structure of the L-ARG competitive inhibitor, compound **9**, was built in a Maestro molecular modelling environment (Maestro release 2016) and minimized using MacroModel software (MacroModel, Schrödinger, LLC, New York, NY, 2016) as previously reported by us [25,28], employing OPLS-AA 2005 as a force field and the GB/SA model for simulating the solvent effects. The PRCG method with 1000 maximum iterations and a 0.001 gradient convergence threshold was employed. Furthermore, the LigPrep (LigPrep, Schrödinger, LLC, New York, NY, 2016) application was used to refine the chemical structure.

*Protein preparation*: According to our previous work [15,19,24,25], we used the recently experimentally solved structure of *L. mexicana* arginase (PDB ID: 4IU1) [29] downloaded from the Protein Data Bank (PDB) and imported into the Maestro molecular modeling environment. The structure was submitted to the protein preparation wizard protocol implemented in Maestro suite 2016 (Protein Preparation Wizard workflow 2016) in order to obtain a reasonable starting structure for further computational experiments.

*Molecular docking*: The docking experiments were performed by Glide (Glide, Schrödinger, LLC, New York, NY, 2016) using the ligand and the protein prepared as mentioned above with the Glide extra precision (XP) method. The energy grid was prepared using the default value of the protein atom scaling factor (1.0 Å) within a cubic box centered on the crystallized ligand nor-NOHA. As part of the grid generation procedure, metal constraints for the receptor grids were also applied [30,31]. After the grid generation, the ligands were docked into the enzymes considering the metal constraints. The number of poses entered into the post-docking minimization was set to 100 and the Glide XP score was evaluated. The XP method was able to correctly accommodate the crystallized inhibitor into the binding site (data not shown).

*Estimated ligand binding energy*: The Prime/MM-GBSA method implemented in Prime software (Prime, Schrödinger, LLC, New York, NY, 2016) computed the change between the free and the complex state of both the ligand and the protein after energy minimization. The technique was used on the docking complex of the compound presented in this study. The software was employed to calculate the ligand binding energy (ΔG_bind_), as previously reported [19,24,25].

*Molecular Dynamics*: Desmond 5.6 academic version via Maestro was used to perform the MD simulation studies (Desmond Molecular Dynamics System, version 5.6, D.E. Shaw Research, New York, NY, 2018. Maestro-Desmond Interoperability Tools, Schrödinger, New York, NY, 2018). The MD simulations were executed on two NVIDIA GPUs, employing the Compute Unified Device Architecture (CUDA) API [32]. The complex resulting from the molecular docking studies was inserted into an orthorhombic box together with water molecules and simulated by the solvent model TIP3P using the Desmond system builder available in Maestro [31,33]. The MD simulation was performed adopting OPLS as the force field [34]. Na^+^ and Cl^−^ ions (0.15 M) were added to mimic the physiological concentration of the monovalent ions. The ensemble class NPT (constant number of particles, pressure, and temperature) was used adopting a constant temperature of 310 K and a pressure of 1.01325 bar. For integrating the equations of motion, a RESPA integrator [35] was used. Nosé–Hoover thermostats [36] and the Martyna–Tobias–Klein method [37] were employed to maintain a constant temperature and pressure of the simulation, respectively. The particle mesh Ewald technique (PME) was employed to calculate the long-range electrostatic interactions [38]. A threshold of 9.0 Å was chosen for the van der Waals and short-range electrostatic interactions. The system was equilibrated using the default procedure, which consisted of a series of restrained minimization and MD simulations to gradually relax the system. As a result, a single 300 ns trajectory was determined. The MD simulation studies were independently repeated two times to provide a more reliable output. Simulation event analysis tools included in the software package were used to examine the trajectory files. All charts relating to the MD simulation presented in this article were created using the same tools. Therefore, the RMSD was evaluated using the following equation:$${\mathrm{RMSD}}_{x}=\sqrt{\frac{1}{N}\sum_{i=1}^{N}{\left({r^{\prime }}_{i}\left(t_{x}\right)-r_{i}\left(t_{ref}\right)\right)}^{2}}$$
where ***RMSD_x_*** is the calculation for a frame ***x***; ***N*** is the number of selected atoms; ***t_ref_*** is the reference time (normally the first frame was utilized as the reference at time ***t*** = 0); and ***r′*** is the position of the chosen atoms in frame ***x*** at time ***t_x_*** after the superimposition with the reference frame. Every frame in the simulation trajectory was subjected to the same technique. The following formula was used to calculate the root mean square fluctuation (***RMSF***):$$RMSF_{i}=\sqrt{\frac{1}{T}\sum_{t=1}^{T}<{\left({r^{\prime }}_{i}\left(t\right)-r_{i}\left(t_{ref}\right)\right)}^{2}>}$$
where ***RMSF_i_*** is the generic residue; ***T*** is the trajectory time considered for the calculation of the ***RMSF***; ***t_ref_*** is the reference time; ***r_i_***^:^ is the position of residue ***i***; and ***r′*** is the position of the atoms in residue ***i*** after the superposition on the reference. The square distance was averaged over the atoms in the residue, as indicated by the angle brackets. 

## 5. Conclusions

In summary, we present a novel biosynthetic approach that increases the possibility of producing L-ARG inhibitors via engineered *E. coli* and opens the possibility of exploring unnatural substrates to produce new compounds with the clinical potential to treat leishmaniasis. In this work, by using a multidisciplinary approach, we reported the production, experimental evaluation of the mechanism of action as L-ARG inhibitors, and an investigation of the inhibitory potential at the molecular level by applying an appropriate computational protocol. Specifically, cinnamic esters and cinnamide-derived compounds revealed an approach to design optimized L-ARG inhibitors, paving the way to develop effective drug candidates for treating leishmaniasis.

## Data Availability

Not applicable.

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
