# Peer review of "In Vitro and In Silico Analyses of New Cinnamid and Rosmarinic Acid-Derived Compounds Biosynthesized in Escherichia coli as Leishmania amazonensis Arginase Inhibitors"

_pathogens, 2022, doi:10.3390/pathogens11091020_

Round 1

Reviewer 1 Report

The present work presents a relevant study on the inhibition of arginase in L. amazonensis by caffeic and rosmarinic acid derivatives. Of these evaluated derivatives, three are new compounds. Molecular docking studies were performed to better rationalize the interactions of compound 9 with the target under study. The work has relevance for publication in Pathogens. However, I request you to add the 13C NMR data to the manuscript and to the "supplementary material" the 13C NMR spectrum and MS spectra of compounds 8-10.

Author Response

Reviewer 1

The present work presents a relevant study on the inhibition of arginase in L. amazonensis by caffeic and rosmarinic acid derivatives. Of these evaluated derivatives, three are new compounds. Molecular docking studies were performed to better rationalize the interactions of compound 9 with the target under study. The work has relevance for publication in Pathogens. However, I request you to add the 13C NMR data to the manuscript and to the "supplementary material" the 13C NMR spectrum and MS spectra of compounds 8-10.

Authors: We thank the reviewer for the positive evaluation of the manuscript. In the revised version we addressed the comments, that allowed us to improve the quality of the paper. The 13C NMR data of compounds 8-10 has been added to the revised manuscript and the 13C NMR spectra and MS spectra of compounds 8-10 have been added to the "supplementary material".

Reviewer 2 Report

This is a very interesting article concerning the bioproduction of new antileishmanial compounds, focus on L. amazonensis arginase inhibition. It is well written and well executed, and of interest to the readership. I only have two comments which I have detailed below.

1) Aiming to improve the scientific quality of the manuscript, the authors should generate in vitro proof-of-concept using cultured parasites, as intracellular amastigotes. Besides, they should calculate the selectivity index of the tested compounds, given by the ratio of the Half Cytotoxic Concentration against the host mammalian cell and the Half Effective Concentration against the parasite.

2) The period on Line 326 seems incomplete. Please revise: The concentration that inhibits 50% of the catalytic activity of the enzyme (IC50).

Author Response

Reviewer 2

This is a very interesting article concerning the bioproduction of new antileishmanial compounds, focus on L. amazonensis arginase inhibition. It is well written and well executed, and of interest to the readership. I only have two comments which I have detailed below.

Authors: We thank the reviewer for the positive evaluation of the manuscript. In the revised version we addressed the comments, that allowed us to improve the quality of the paper.

1) Aiming to improve the scientific quality of the manuscript, the authors should generate in vitro proof-of-concept using cultured parasites, as intracellular amastigotes. Besides, they should calculate the selectivity index of the tested compounds, given by the ratio of the Half Cytotoxic Concentration against the host mammalian cell and the Half Effective Concentration against the parasite.

Authors: The main objective of this study was to verify the inhibitory activity of parasite arginase and in silico study for the special issue. These results open a possibility of further study including intracellular amastigotes and citotoxity.

2) The period on Line 326 seems incomplete. Please revise: The concentration that inhibits 50% of the catalytic activity of the enzyme (IC50).

Authors: We thank the reviewer for his/her careful reading. According to the comments, we modified the sentence at line 326-327. Now the resulting sentence is as follows: The concentration that inhibits 50% of the catalytic activity of the enzyme (IC50) was determined using inhibitor concentration from 200 μM to 0.05 μM; the concentrations were obtained via serial dilution in water (1:4).

Reviewer 3 Report

In the current manuscript, the authors presented a study on new cinnamide and rosmarinic acid-derived compounds biosynthesized in E. coli as L. amazonensis arginase inhibitors. The study is very interesting and important because it may contribute to the development of an effective drug against leishmaniasis.

The investigation was well planned and in general well described in the manuscript. Nevertheless, I have a few following points that should be considered/corrected:

1. Please provide in the introduction (lines 38-42) more details about Leishmania spp. and leishmaniasis and the need to find new drugs against this disease.

2. Please consider including Figures 4-6 and refer to them in the results section rather than in the discussion section.

3. Please adjust Figure 6 to fit on the page.

4. Table 1: Please explain all abbreviations used in the table.

Author Response

Reviewer 3

In the current manuscript, the authors presented a study on new cinnamide and rosmarinic acid-derived compounds biosynthesized in E. coli as L. amazonensis arginase inhibitors. The study is very interesting and important because it may contribute to the development of an effective drug against leishmaniasis.

The investigation was well planned and in general well described in the manuscript. Nevertheless, I have a few following points that should be considered/corrected:

Authors: We thank the reviewer for the positive evaluation of the manuscript. In the revised version we addressed the comments, that allowed us to improve the quality of the paper.

  1. Please provide in the introduction (lines 38-42) more details about Leishmania spp. and leishmaniasis and the need to find new drugs against this disease.

Authors: We thank the reviewer for the suggestion. We added details about leishmaniasis and insurgence of resistance to the classical treatments, making necessary the identification of novel drug candidates with a new reference (at line 43-46, now reading “The most antileishmanial agents are high toxicity injectable medicines such as pentavalent antimonials (sodium stibogluconate and meglumine antimoniate) and liposomal amphotericin B. The resistance of these drugs was also described for both classes as well to miltefosine, an oral medicine used to treat leishmaniasis [2]).

  1. Please consider including Figures 4-6 and refer to them in the results section rather than in the discussion section.

Authors: According to the reviewer suggestion, we included Figures 4-6 with related findings in the Results sections.

  1. Please adjust Figure 6 to fit on the page.

Authors: Figure 6 was adjusted to fit on the page.

  1. Table 1: Please explain all abbreviations used in the table.

Authors: abbreviations included (MW = Molecular Weight; IC50 = concentration that inhibits 50% of enzyme activity. Ki = inhibition constant).

Round 2

Reviewer 3 Report

I am fully satisfied with the authors' answers and the changes in the manuscript. The revised manuscript is very well prepared. I have no critical comments and in my opinion the manuscript is suitable for acceptance and publication.

Author Response

I am fully satisfied with the authors' answers and the changes in the manuscript. The revised manuscript is very well prepared. I have no critical comments and in my opinion the manuscript is suitable for acceptance and publication.

Authors: we thank the reviewer for the positive evaluation of the revised version of the manuscript.